# Efficacy Comparison Study of Human Epidermal Growth Factor (EGF) between Heberprot-P^®^ and Easyef^®^ in Adult Zebrafish and Embryo under Presence or Absence Combination of Diabetic Condition and Hyperlipidemia to Mimic Elderly Patients

**DOI:** 10.3390/geriatrics7020045

**Published:** 2022-04-06

**Authors:** Kyung-Hyun Cho, Ju-Hyun Kim, Hyo-Seon Nam, Dae-Jin Kang

**Affiliations:** 1Raydel Research Institute, Medical Innovation Complex, Daegu 41061, Korea; aksk1694@rainbownature.com (J.-H.K.); sun91120@rainbownature.com (H.-S.N.); daejin@rainbownature.com (D.-J.K.); 2LipoLab, Yeungnam University, Gyeongsan 38541, Korea

**Keywords:** epidermal growth factor, Heberprot-P75^®^, Easyef^®^, zebrafish, embryo, wound healing

## Abstract

Recombinant human epidermal growth factor (EGF) has been used to treat adult diabetic foot ulcers and pediatric burns by facilitating wound healing and epithelization, especially for elderly patients. Several formulation types of EGF from different expression hosts are clinically available, such as intralesional injection and topical application. On the other hand, no study has compared the in vivo efficacy of EGF products directly in terms of tissue regeneration and wound healing activity. The present study compared two commercial products, Heberprot-P75^®^ and Easyef^®^, in terms of their tissue regeneration activity in adult zebrafish and the developmental speed of zebrafish embryos. Fluorescence spectroscopy showed that the two EGF products had different Trp fluorescence emission spectra: Easyef^®^ showed a wavelength of maximum fluorescence (WMF) of 337 nm with weak fluorescence intensity (FI), while Heberprot-P75^®^ showed WMF of 349 nm with a 4.1 times stronger FI than that of Easyef^®^. The WMF of Heberprot-P75^®^ was quenched by adding singlet oxygen in ozonated oil, while the WMF of Easyef^®^ was not quenched. Treatment with Heberprot-P75^®^ induced greater embryo development speed with a higher survival rate after exposure to EGF in water and microinjection into embryos. Under normal diet (ND) consumption, Heberprot-P75^®^ showed a 1.4 times higher tail fin regeneration activity than Easyef^®^ during seven days from the intraperitoneal injection (10 μL, 50 μg/mL) after amputating the tail fin. Under ND consumption and diabetic condition caused by streptozotocin (STZ) treatment, Heberprot-P75^®^ showed 2.1 times higher tail fin regeneration activity than Easyef^®^ from the same injection and amputation protocol. Under a high-cholesterol diet (HCD) alone, Heberprot-P75^®^ showed 1.2 times higher tail fin regeneration activity than the Easyef^®^ group and PBS group from the same injection and amputation. Under diabetic conditions (STZ-injected) and HCD consumption, the Heberprot-P75^®^ group showed 1.7 and 1.5 times higher tail fin regeneration activity than the Easyef^®^ group and PBS group, respectively, with a distinct and clean regeneration pattern. In contrast, the Easyef^®^ group and PBS group showed ambiguous regeneration patterns with a severe fissure of the tail fin, which is a typical symptom of a diabetic foot. In conclusion, Heberprot-P75^®^ and Easyef^®^ have different Trp fluorescence properties in terms of the WMF and fluorescence quenching. Treatment of Heberprot-P75^®^ induced a greater developmental speed of zebrafish embryos in both water exposure and microinjection. Heberprot-P75^®^ induced significantly higher wound healing and tissue regeneration activity than Easyef^®^ and PBS in the presence or absence of diabetic conditions and cholesterol supplementation.

## 1. Introduction

Epidermal growth factor (EGF), a glycosylated polypeptide with 53 amino acids, has been used widely for medical treatment [1] and regenerative medicine [2,3] because it facilitates epidermal cell replication, differentiation, and proliferation [4]. Several EGF products with several formulae, such as intralesional injection, topical spray, dermal cream, and ointment, are commercially available to treat or alleviate diabetic foot ulcers (DFUs) [5,6] and provide anti-aging skincare [7]. DFUs are severe complications of diabetes mellitus and a worldwide health burden [8,9], frequently leading to painful amputations. Among the diabetic complications, 15–20% of diabetic patients will develop foot ulcers during their disease with severe impairment of the quality of life [10]. Especially for elderly patients, DFUs should be treated as early as possible.

There have been many commercial products of EGF to treat DFUs via wound healing and tissue regeneration. Heberprot-P^®^, a lyophilized powder for injection, was constituted with expressed and purified EGF from the yeast system (*Saccharomyces cerevisiae*). Heberprot-P^®^ was reported to have wound healing efficacy for DFUs by intralesional injections [11]. Easyef^®^, which is a water spray formula containing purified EGF from a bacterial system (*Escherichia coli*), has positive effects on the healing of DFUs via topical applications [12,13]. Although the efficacy and cost are controversial, several local delivery methods to treat DFUs, e.g., intralesional injection (Heberprot-P^®^) and topical administration, have been reported [14].

Despite the controversial data, no study has compared the two products directly using an in vitro and in vivo animal model. In the current report, adult zebrafish and its embryo model were used to compare tail fin regeneration speed and developmental speed of embryos. Zebrafish (*Danio rerio*) is a widely used vertebrate model in the study of tissue regeneration [15] because of its high regenerative capacity and physiological similarities to humans. Zebrafish respond efficiently to diabetic conditions induced by a streptozotocin (STZ) injection [16]. Hypercholesterolemia can be induced easily in zebrafish by a high-cholesterol diet (HCD) [17,18]. One feature of metabolic syndrome, decreased HDL-cholesterol, is associated with a high risk of diabetic feet [19], indicating that dyslipidemia is a risk factor of DFUs. Zebrafish were treated with an STZ injection and cholesterol supplementation to mimic hyperlipidemia and diabetic foot complications in humans. Zebrafish embryos can be an excellent model to monitor the developmental speed because zebrafish embryos develop externally. In addition, because zebrafish embryos are optically transparent during development, both water exposure and microinjection into the embryo are possible, which allow a comparison of the developmental speed and morphology [20].

Because the primary sequence of EGF showed two Trp residues at the 49th and 50th amino acids in the C-terminal region, it is possible to compare wavelength maximum fluorescence (WMF) between Heberprot-P^®^ and Easyef^®^ as well as fluorescence intensity. The exposure extent of Trp to the hydrophilic phase in tertiary structure might be associated with peptide activity. However, there have been no study to directly compare its in vivo efficacy and in vitro characterization between Heberprot-P^®^ and Easyef^®^. The comparison result would be useful for elderly patients to enable them to choose commercially available EGF products.

The current study was designed to compare the in vivo efficacy of two EGF products in tissue regeneration and wound healing activity. Heberprot-P^®^ and Easyef^®^ were injected individually into the intraperitoneal space of adult zebrafish, in the absence or presence of both diabetic condition and cholesterol supplementation to mimic hyperlipidemia observed in the elderly. Zebrafish embryos were also treated with Heberprot-P^®^ and Easyef^®^ to compare the influence of water exposure and microinjection on developmental speed and morphology. Heberprot-P^®^ and Easyef^®^ were compared based on the spectroscopic properties regarding intrinsic Trp fluorescence, and collisional quenching was used for in vitro comparison. The in vitro and in vivo evaluations of the two EGF pharmaceuticals can provide useful information about better choice for elderly patients.

## 2. Materials and Methods

### 2.1. Materials

Heberprot-P75^®^, recombinant human EGF expressed and purified from yeast expression system [21], in a lyophilized powder formula (Lot # 1701Z1/3), was obtained from the Center for Genetic Engineering and Biotechnology (CIGB, Havana, Cuba). Easyef^®^ (Daewoong Pharmaceuticals, Seoul, Korea) in spray water formula (Lot # B01613, 0.05 mg/mL of EGF and 2 mg/mL of methyl-parabenzoic acid as a preservative) was purchased from a local hospital under prescription. Heberprot-P75^®^ was dissolved in distilled water (final 0.05 mg/mL) and stored at 4 °C without the addition of preservatives. The vortexed and mixed form of Easyef^®^, which was freshly opened prior to use, was also stored at 4 °C. Methyl-parabenzoic acid (CAS-No. 99-76-3, Sigma-Aldrich Cat # H5501) was purchased from Sigma-Aldrich Co. (St. Louis, MO, USA).

### 2.2. Fluorescence Spectroscopy

The primary sequence of EGF showed two Trp residues at the 49th and 50th amino acids in the C-terminal region. The Trp fluorescence was measured by determining the emission fluorescence maxima from the uncorrected spectra obtained on an FL6500 spectrofluorometer (Perkin-Elmer, Norwalk, CT, USA) using Spectrum FL software version 1.2.0.583 (Perkin-Elmer) using a 1 cm path-length Suprasil quartz cuvette (Fisher Scientific, Pittsburgh, PA, USA). The three equally diluted samples, i.e., Heberprot-P75^®^, Easyef^®^, and methyl-parabenzoic acid, were excited at 295 nm to avoid tyrosine fluorescence. The emission spectra were scanned from 190 to 900 nm at room temperature.

### 2.3. Zebrafish and Embryo

The zebrafish and embryos were maintained using the standard protocols [22]. The maintenance of zebrafish and the procedures using zebrafish were approved by the Committee of Animal Care and Use of Raydel Research Institute (approval code RRI-20-003). The fish were maintained in a system cage at 28 °C under a 10:14 h light cycle with the consumption of normal tetrabit (Tetrabit Gmbh D49304, 47.5% crude protein, 6.5% crude fat, 2.0% crude fiber, 10.5% crude ash), containing vitamin A (29,770 IU/kg), vitamin D3 (1860 IU/kg), vitamin E (200 mg/kg), and vitamin C (137 mg/kg), Melle, Germany.

### 2.4. Water Exposure of Zebrafish Embryos

Distilled water containing the embryos at one-hour post-fertilization (hpf) was treated with equally diluted 10 μL of (final 0.5 μg/mL) Heberprot-P75^®^ and Easyef^®^. The embryos were incubated at 28 °C and the developmental stage was monitored for 72 h using a stereomicroscope (Motic SMZ 168; Hong Kong).

### 2.5. Microinjection of Zebrafish Embryos

Embryos at one hpf were injected individually by a microinjection using a pneumatic picopump (PV830; World Precision Instruments, Sarasota, FL, USA) equipped with a magnetic manipulator (MM33; Kantec, Bensenville, IL, USA) and a pulled microcapillary-pipette-using device (PC-10; Narishigen, Tokyo, Japan). The bias was minimized by performing the injections at the same position on the yolk. After the same volume (4 nL) of each peptide injection (200 pg/4 nL), live embryos were observed under a stereomicroscope (Motic SMZ 168; Hong Kong) and photographed using a Motic cam2300 CCD camera for 72 h.

### 2.6. Tail Fin Regeneration

The wound-healing effect of the EGF peptides was tested using adult zebrafish. For the fin regeneration studies, zebrafish were anesthetized by submersion in 2-phenoxyethanol (Sigma P1126; St. Louis, MO, USA) in system water (1:1000 dilution). Tail fins of the zebrafish, approximately 24 weeks old, were cut with a scalpel close to the proximal branch point of the dermal rays within the fin.

The subcutaneous injection of streptozotocin (STZ) at the nearby abdomen region was carried out using a 26-gauge needle microsyringe (SGE, Ringwood, Australia) to deliver 30 µL of 0.3% STZ (Sigma S0130; Sigma) in 5 mM citrate buffer. Before being injected five times with STZ over eight days, as shown in Figure 1, the experimental zebrafish group consumed two diet groups: one group consumed normal tetrabit (normal diet group, n = 200), and the other group consumed tetrabit containing 4% cholesterol (high-cholesterol diet (HCD) group, n = 200).

After four weeks of consumption of the designated diet with five STZ injections, at day 0, Heberprot-P75^®^ or Easyef^®^ was injected intraperitoneally with tail fin amputation. For the fin regeneration studies, the diabetic and non-diabetic zebrafish were anesthetized and the tail fins were cut with a scalpel close to the proximal branch point of the dermal rays within the fin.

After amputation, 10 µL of each peptide (50 µg/mL of protein) was injected intraperitoneally near the lower abdomen. After the injection, the fish were maintained in a 28 °C incubator. Images of the regenerating fins from the live zebrafish were taken at one-day intervals, until seven days, under a stereomicroscope (Motic SMZ 168; Hong Kong) and photographed using a Motic cam2300 CCD camera, with Image Proplus software version 4.5.1.22 (Media Cybernetics, Bethesda, MD, USA).

### 2.7. Experimental Design of Tail Fin Regeneration

For the experimental design of tail fin regeneration, as shown in Figure 1, four sets of combinations with a high-cholesterol diet (HCD) and STZ injection were carried out to compare the efficacy of Heberprot-P75^®^ and Easyef^®^.

### 2.8. Statistical Analysis

The data of this study are expressed as the mean ± SD from at least three independent experiments with duplicate samples. For the zebrafish study, multiple groups were compared by a one-way analysis of variance (ANOVA) using the Scheffe test. Statistical analysis was performed using the SPSS software program (version 23.0; SPSS, Inc., Chicago, IL, USA). A *p*-value < 0.05 was considered statistically significant.

## 3. Results and Discussion

### 3.1. Comparison of the Fluorescence Emission Spectra

From the 53-amino-acid sequence, human EGF has a slightly acidic isoelectric point 4.6–4.9 with a molecular weight of 6045 based on the calculation [23]. The primary sequence of the EGF contains two bis Trp residues at 49th and 50th position and five Tyr residues at 3rd, 10th, 13th, 29th, and 37th position without Phe. We compared Trp fluorescence intensity and location of Trp in the three-dimensional structure of the pepide via measurement of wavelength maximum fluorescence (WMF) between the Heberprot-P75^®^ and Easyef^®^.

At the same concentration of peptide (0.05 mg/mL), the full range of pre-scan emissions (190–900 nm) revealed that Heberprot-P75^®^ had two maxima at 298.7 and 599.5 nm, while Easyef^®^ showed maxima at 327.5 and 653.7 nm, as shown in Figure 2A, indicating that UV-region fluorescence was redshifted by 28.8 nm in Easyef^®^. Interestingly, the emission spectrum of methyl-parabenzoic acid showed two maxima at 325.5 and 650.9 nm, which is a very similar spectrum pattern with Easyef^®^. This result suggests that the fluorescence spectrum of Easyef^®^ might have originated, at least in part, from methyl-parabenzoic acid because the Easyef^®^ formula contained higher content of methyl-parabenzoic acid (final 2 mg/mL) than recombinant hEGF (0.05 mg/mL).

As shown in Figure 2B, wavelength maximum fluorescence (WMF) of Trp at Ex = 295 nm (Em = 305–400 nm) to avoid Tyr excitation showed that Heberprot-P75^®^ and Easyef^®^ had a WMF of 349 nm (intensity = 202,356) and 337 nm (intensity = 49,008), respectively, suggesting that the WMF of Heberprot-P75^®^ showed a redshift with an approximately 4.1 times stronger fluorescence intensity (FI) than Easyef^®^. Interestingly, methyl-parabenzoic acid showed a similar pattern of WMF around 338 nm with Easyef^®^, even though the FI was much smaller, approximately 14,072. Overall, the two Trp residues in Heberprot-P75^®^ showed greater redshift in the WMF when exposed to the aqueous phase compared to that of Easyef^®^, indicating that the secondary structure was more open to the hydrophilic phase.

Ozonated sunflower oil (OSO) was treated to each peptide solution (final 1%, 2%, and 4% of OSO) to compare the change of the intrinsic fluorescence by collisional quenching with singlet oxygen as described elsewhere [24]. A putative oxygen species or singlet oxygen (^1^O_2_) in OSO causes the collisional quenching of the indole side chain of Trp fluorescence [25]. Excited-state oxygen (^1^O_2_) quenches the fluorescence of an organic chromophore at the diffusion-controlled limit [26]. Singlet oxygen is also susceptible to removal by Trp, resulting in fluorescence bleaching—a decrease in the intensity of Trp fluorescence [24,26]. From the same WMF measurement, the fluorescence intensity (FI) decreased sharply due to the OSO treatment (final 4%) in high-density lipoproteins (HDL_3_), up to a 75% decrease in FI from the initial level [24]. In the same context, the Trp intensity of Heberprot-P75^®^ was quenched by adding the OSO in a dose-dependent manner up to 30% decrease of FI by the OSO treatment (final 4%). In contrast, the FI of Easyef^®^ was not quenched by the same addition of OSO (inset graph of Figure 2B). Interestingly, the FI of methyl-parabenzoic acid was not quenched by the same OSO addition (inset graph of Figure 2B). These results suggest that the fluorescence of Heberprot-P75^®^ originated from the intrinsic fluorescence of the two Trp residues, but the fluorescence of Easyef^®^ might not. During quenching, the WMF of Heberprot-P75^®^ was not changed around 349 nm, even though the FI was remarkably decreased, indicating that the two Trp residues were fully exposed to the hydrophilic phase. Similarly, Victorovich et al. [27] reported that human EGF showed two emission fluorescence peaks at 331 and 357 nm at Ex = 280 nm. Despite the possibility of Tyr excitation (Ex = 280 nm), these results show a good agreement with current results that human EGF exhibited redshifted fluorescence around 350–360 nm.

### 3.2. Developmental Speed after Water Exposure

In order to compare peptide activity in vivo, we compared developmental speed and hatching ratio in zebrafish embryo system. After the water exposure of peptides, the Heberprot-P75^®^ (final 0.5 μg/mL) exposed embryo (n = 50) showed an approximately 15% higher hatching ratio (*p* < 0.05), and greater developmental speed than the Easyef^®^ (final 0.5 μg/mL) exposed embryo (n = 50), as shown in Figure 3. At 30 h, the Heberprot-P75^®^ group showed the greatest developmental speed with darker pigmentation in the eye and good development of notochord, indicating that the Heberprot-P75^®^ group entered into the pharyngula period, while the Easyef^®^ and PBS groups were still in the segmentation period. At 60 h, the Heberprot-P75^®^ group showed a typical pattern of after pec-fin stage darker pigment in eye and notochord, whereas the Easyef^®^ and PBS groups were in the long-pec stage.

This result suggests that the recombinant human EGF could facilitate the faster development of zebrafish embryos by diffusion with water exposure. Although the EGF network contributes to the oocyte developmental quality and healthy embryo development in animal and human models [28], the current result proved that human EGF could accelerate the development of zebrafish embryos. Sargent et al. reported that treatment of human embryos with heparin-binding EGF promoted the developmental speed of the embryo [29]. Overall, EGF treatment could enhance the embryo developmental quality and survivability in humans and zebrafish.

### 3.3. Embryo Development after Microinjection

For the direct delivery of the peptide into embryos, a microinjection was carried out inside the embryo yolk at 1 h post-fertilization. At day one post-injection, the Heberprot-P75^®^-injected embryo showed darker pigmentation in the eye and longer notochord and tail around the twenty-five-somite stage. In contrast, the Easyef^®^- and PBS-injected embryos showed a lower speed between the seventeen-somite stage and the twenty-somite stage (Figure 4). At day two post-injection, the Heberprot-P75^®^-injected embryo (n = 60) showed higher survivability and a greater developmental speed than that of Easyef^®^ (n = 60) with 73% and 45% survivability, respectively, as shown in Figure 4A. The Heberprot-P75^®^-injected embryo entered the high-pec stage, displaying a well-developed notochord and stand-up posture, but the Easyef^®^-injected and PBS-injected embryos showed a reduced development speed in the primordium stage.

### 3.4. Tail Fin Regeneration under ND Consumption Alone

The first set (Figure 5), under the ND-alone consumption, showed that the Heberprot-P75^®^ group (7.624 ± 0.571 mm^2^) had a 1.4 times greater regeneration area than the Easyef^®^ group (5.402 ± 0.496 mm^2^, *p* = 0.018) and PBS (*p* = 0.011) group (5.333 ± 0.540 mm^2^, *p* = 0.011) during the seven days post-injection. No fissure in the tail was found in any of the groups during the seven days. On day three, both peptides showed an approximately 2.2 times higher tissue regeneration speed than PBS (1.856 ± 0.363 mm^2^), but there was no difference between Heberprot-P75^®^ (4.181 ± 0.408 mm^2^) and Easyef^®^ (4.219 ± 0.415 mm^2^). On the other hand, the Easyef^®^-injected group showed a similar regeneration area to that of the PBS-injected group on day seven.

### 3.5. Tail Fin Regeneration under Diabetic Condition and ND Consumption

In the second set, as shown in Figure 6A, for ND consumption under diabetic conditions (STZ-injected), Heberprot-P75^®^ showed the highest regeneration speed during days three, five, and seven. On day seven, the Heberprot-P75^®^-injected zebrafish (4.113 ± 0.643 mm^2^) showed 2.1 times (*p* = 0.033) and 1.5 times (*p* = 0.025) higher regeneration speed than the Easyef^®^-injected (1.994 ± 0.375 mm^2^) and PBS-injected zebrafish (2.849 ± 0.455 mm^2^), respectively. Interestingly, the Easyef^®^ group showed the lowest regeneration speed with several fissures of the tail fin followed by the PBS group (Figure 6B).

The fissure appeared on day three in the Easyef^®^ group (n = 2) and PBS group (n = 4), while the Heberprot-P75^®^ group showed no fissure during the seven days. On day five, the Heberprot-P75^®^-injected zebrafish (3.022 ± 0.491 mm^2^) showed 1.9 times (*p* = 0.044) higher regeneration speed than the Easyef^®^-injected zebrafish (1.994 ± 0.375 mm^2^). Although there was no significance, the Heberprot-P75^®^-injected zebrafish showed 1.4 times (*p* = 0.311) higher regeneration speed than the PBS-injected zebrafish (2.092 ± 0.283 mm^2^).

### 3.6. Tail Fin Regeneration under HCD Consumption Alone

In the third set, HCD consumption alone (Figure 7), the Heberprot-P75^®^ injection showed the highest regeneration speed during the seven days, up to 9.888 ± 0.366 mm^2^, while the Easyef^®^ and PBS injection groups showed regeneration speeds of 8.091 ± 0.556 mm^2^ (*p* = 0.048) and 8.176 ± 0.296 mm^2^ (*p* = 0.005), respectively. Heberprot-P75^®^-injected zebrafish showed a 1.2 times greater regeneration area than the Easyef^®^-injected and PBS-injected zebrafish at day seven. No fissure in the tail was found in all groups during the seven days.

Until five days, there was no significant difference between the Heberprot-P75^®^ and Easyef^®^ results, 8.308 ± 0.292 mm^2^ and 7.680 ± 0.491 mm^2^ (*p* = 0.552), respectively. On the other hand, Heberprot-P75^®^-injected zebrafish showed 1.3 times (*p* < 0.001) higher regeneration speed than the PBS-injected zebrafish (6.461 ± 0.380 mm^2^). These results suggest that Easyef^®^ showed a more negligible stimulation effect than Heberprot-P75^®^, especially in the long-term effect in both ND-alone and HCD-alone supplementation without the diabetic conditions.

### 3.7. Tail Fin Regeneration under Diabetic Condition and HCD Consumption

In the fourth set, as shown in Figure 8, HCD consumption under diabetic conditions (STZ-injected), Heberprot-P75^®^ also showed the highest regeneration speed during days three, five, and seven. At day seven, the Heberprot-P75^®^-injected zebrafish (5.998 ± 0.687 mm^2^) showed 1.7 times (*p* = 0.047) and 1.6 times (*p* = 0.026) higher regeneration speed than Easyef^®^-injected (3.456 ± 0.541 mm^2^) and PBS-injected (3.838 ± 0.445 mm^2^) zebrafish, respectively. On day five, Heberprot-P75^®^ (4.052 ± 0.485 mm^2^) showed a 1.6 times (*p* = 0.039) and 1.3 times (*p* = 0.242) greater regeneration area than the Easyef^®^-injected (2.564 ± 0.232 mm^2^) and PBS-injected (3.026 ± 0.315 mm^2^) zebrafish, respectively. The first fissure was detected on day 3 in the Easyef^®^ group (n = 1) and PBS group (n = 6), while the Heberprot-P75^®^ group did not show any fissure in the tail fin. Similar to the second set, the Easyef^®^-injected zebrafish showed the lowest regeneration speed with several fissures of the tail fin followed by the PBS-injected zebrafish. These results show that Easyef^®^ induced the lowest regeneration speed under diabetic conditions, even though more fissures were detected in the PBS-injected group.

In the absence of a diabetic condition, the HCD-alone group (9.888 ± 0.366 mm^2^) showed a 1.3 times greater regeneration area than the ND-alone group (7.624 ± 0.571 mm^2^) in the Heberprot-P75^®^-injected group, indicating that cholesterol consumption increased the wound-healing activity. On the other hand, under diabetic conditions, the HCD + STZ group (5.998 ± 0.687mm^2^) showed a 1.4 times greater regeneration area than the ND + STZ group (4.113 ± 0.643 mm^2^), indicating that the tissue regeneration ability was impaired by STZ-injection. Regardless of ND or HCD consumption, the STZ injection caused more fissure of the tail fin in the Easyef^®^ group (30% and 25% of zebrafish in the ND and HCD group, respectively) and PBS group (25% and 20% of zebrafish in ND and HCD group, respectively). Although the reason for this is unclear, the more severe fissure of the tail fin in the Easyef^®^ group than in the PBS group might be due to methyl-parabenzoic acid, a preservative. Fissure of the tail fin is a typical damage pattern of zebrafish with diabetes induced by STZ injection, as described previously [16]. On the other hand, both in the ND + STZ and HCD + STZ groups, Heberprot-P75^®^ injection caused no fissure of the tail fin at all, indicating that it had no toxic or side effects during tissue regeneration, even in the diabetic condition.

As shown in Figure 9, the Heberprot-P75^®^ group showed the highest regeneration area over seven days regardless of the ND or HCD diet and diabetic condition. Among the four combinations, the HCD-alone group showed the greatest regeneration area by an injection of Heberprot-P75^®^, whereas the ND + STZ group showed the least regeneration area. These results showed that cholesterol feeding facilitated tissue regeneration, while the diabetic condition induced by the STZ treatment impaired the regeneration ability. Interestingly, the Easyef^®^ group showed a similar regeneration area to the PBS group in ND-alone and HCD-alone supplementation. On the other hand, the Easyef^®^ group showed a smaller regeneration area than the PBS group in ND + STZ and HCD + STZ, indicating that Easyef^®^ might be less effective under diabetic conditions. To the best of the authors’ knowledge, this is the first report showing that human recombinant EGF (Heberprot-P75^®^) was also effective in diabetic zebrafish, indicating that more application would be possible to other vertebrates. A previous report also showed that the wound healing effect of recombinant human EGF (Easyef^®^) was reproduced in diabetic rats by topical application [30].

Many previous reports showed that the efficacy of EGF was dependent on various concentrations and administration routes (topical application and intralesional injection). An intralesional injection achieves better availability to the deep wound layers, but pain at the injection site is a common complaint of patients. The current results showed that Heberprot-P75^®^, which was purified from the yeast expression system, showed better tissue regeneration efficacy than Easyef^®^, which was purified from the *E. coli* expression system. More glycosylation of EGF from the yeast expression system might contribute to its superior efficacy than the non-glycosylated EGF from the *E. coli* expression system.

## 4. Conclusions

Heberprot-P75^®^ and Easyef^®^ displayed different fluorescence emission spectra. Easyef^®^ showed greater redshift and smaller FI than Heberprot-P75^®^. Water exposure and microinjection into the embryo showed that Heberprot-P75^®^ induced a greater developmental speed with a higher hatching ratio than the Easyef^®^ group. The Heberprot-P75^®^-injected zebrafish showed the highest speed of tail fin regeneration and wound-healing effect in the presence or absence of diabetes with ND or HCD consumption. These results suggest that Heberprot-P75^®^ and Easyef^®^ showed significantly different tissue regeneration and wound healing activities.

## Figures and Tables

**Figure 1 geriatrics-07-00045-f001:**
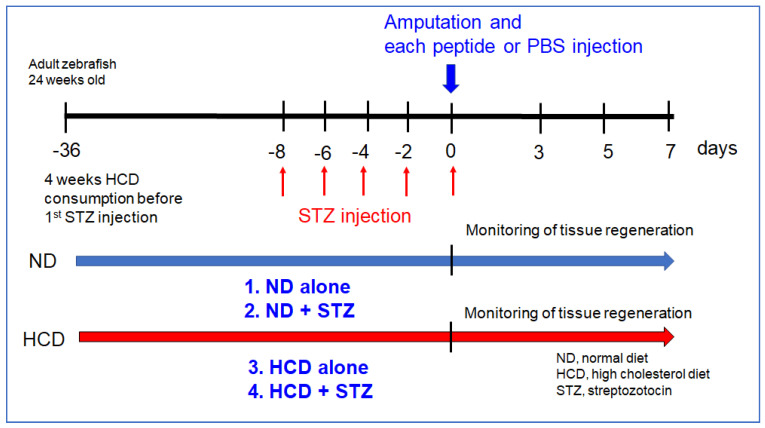
Experimental design of tail fin regeneration in the absence or presence of diabetic condition and hypercholesterolemia induced by 0.3% streptozotocin (STZ) injection and 4% cholesterol feeding. Four types of combination experiments were carried out to monitor the regeneration pattern of the tail fin over seven days.

**Figure 2 geriatrics-07-00045-f002:**
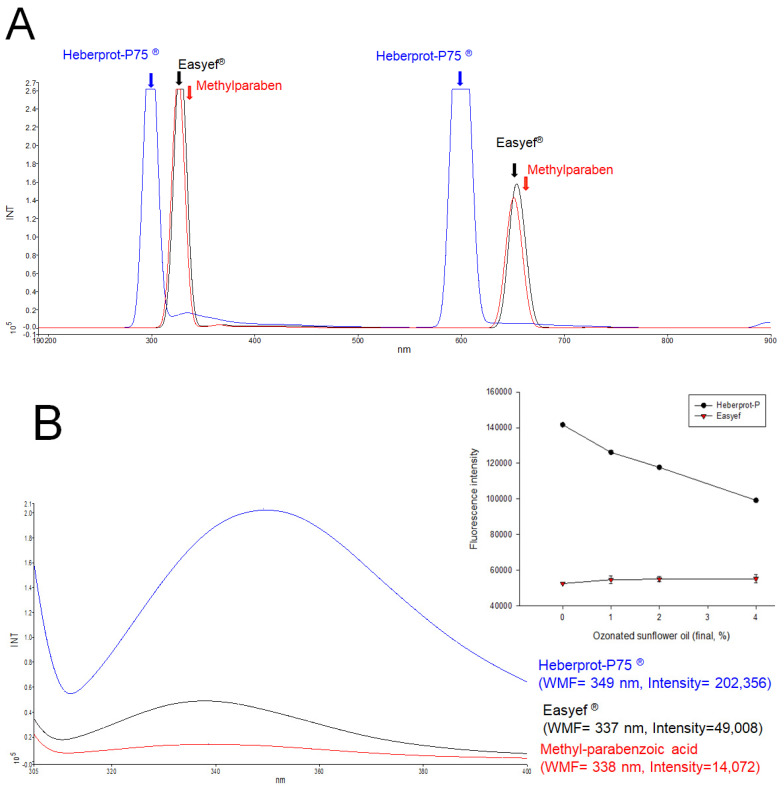
Fluorescence emission spectra of Heberprot-P75^®^, Easyef^®^, and methyl-parabenzoic acid. (**A**) Pre-scan of full-range emission fluorescence spectrum 190–900 nm. (**B**) Wavelength maximum fluorescence (WMF) of Trp spectrum (Ex = 295 nm, Em = 305–400 nm). The inset graph shows the change in the fluorescence intensity depending on ozonated sunflower oil (OSO) treatment. In order to compare fluorophore properties, collisional quenching of Trp was performed by adding the OSO into Heberprot-P75^®^ and Easyef^®^.

**Figure 3 geriatrics-07-00045-f003:**
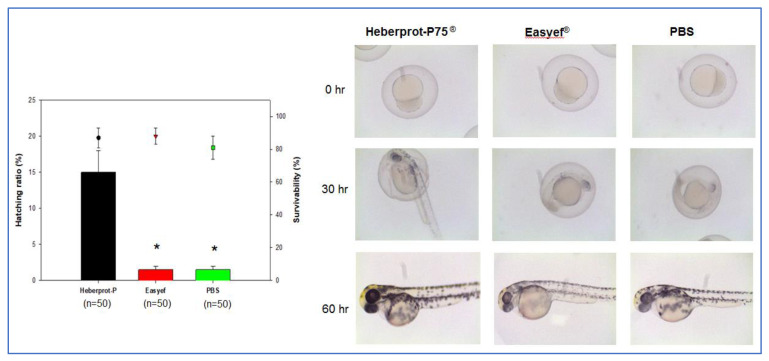
Exposure of Heberprot-P75^®^ and Easyef^®^ (final 0.5 μg/mL) into water containing zebrafish embryo at 1 hpf. The graph showed the hatching ratio and survivability at 30 h post-exposure. *, *p* < 0.05 versus Heberprot-P75^®^. The photographs show the stereo image of embryos from each group at 0, 30, and 60 h post-exposure to compare the developmental stage.

**Figure 4 geriatrics-07-00045-f004:**
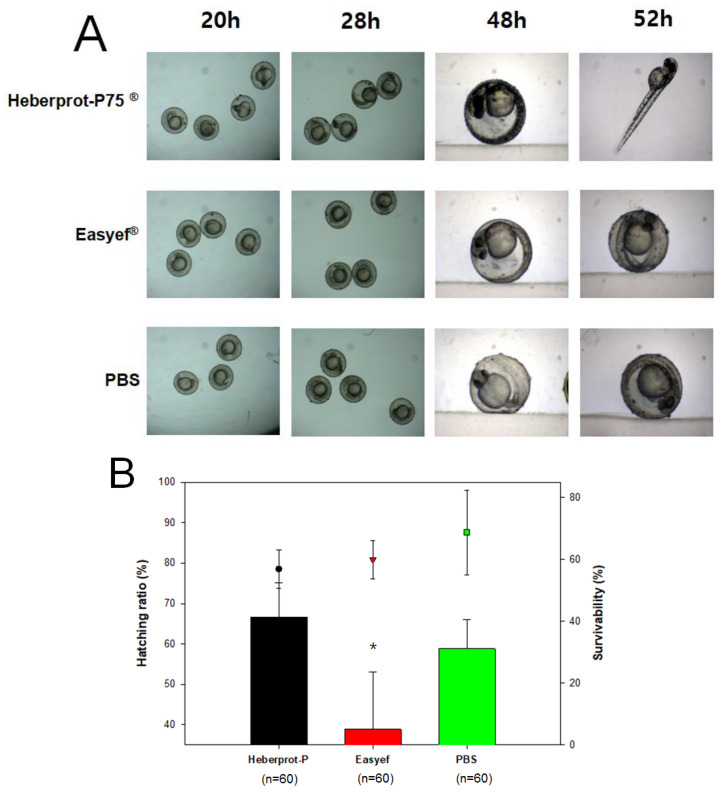
Comparison of developmental speed and morphology after microinjection of Heberprot-P75^®^, Easyef^®^, and PBS. (**A**) Photographs show stereo image of the embryo from each group at 20, 28, 48, and 52 h post-injection to compare the developmental stage. (**B**) The graph shows the hatching ratio and survivability at two days post-exposure. *, *p* < 0.05 versus Heberprot-P75^®^.

**Figure 5 geriatrics-07-00045-f005:**
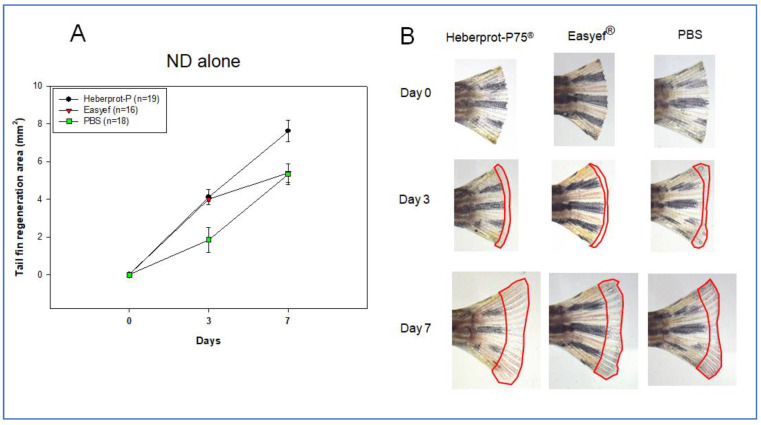
Comparison of tissue regeneration speed (**A**) and morphology in the tail fin (**B**) area under ND consumption during seven days. ND, normal diet.

**Figure 6 geriatrics-07-00045-f006:**
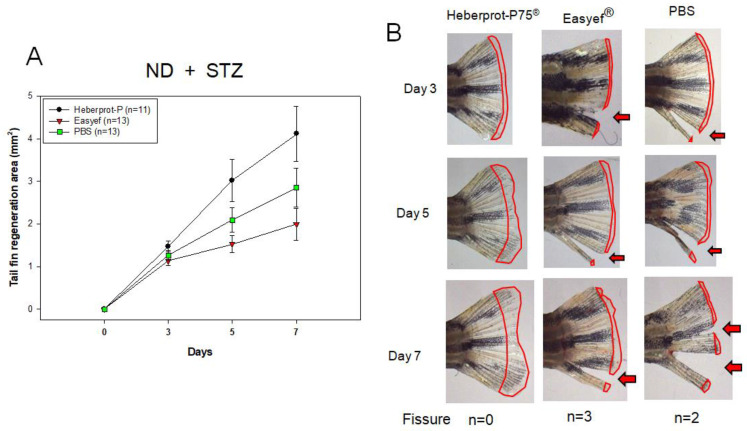
Comparison of tissue regeneration speed (**A**) and morphology (**B**) in the tail fin area under diabetic conditions and ND consumption over seven days. The red arrow indicated the fissure of the tail. At the bottom of the photos, n indicates the number of zebrafish that showed fissure of the tail fin on day 7. ND, normal diet.

**Figure 7 geriatrics-07-00045-f007:**
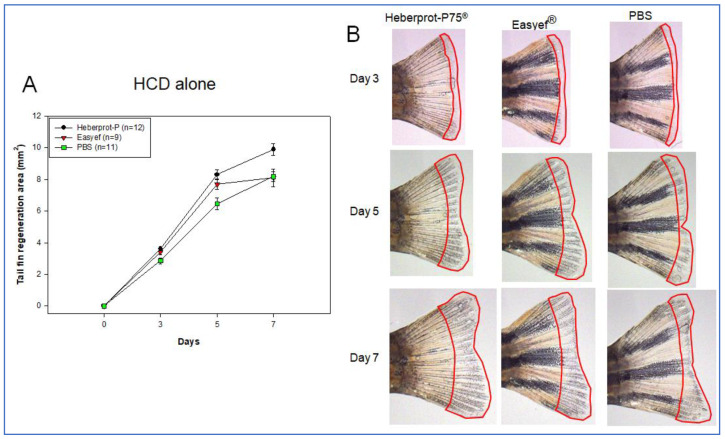
Comparison of tissue regeneration speed (**A**) and morphology in the tail fin area (**B**) under HCD consumption alone over seven days. HCD, high cholesterol diet.

**Figure 8 geriatrics-07-00045-f008:**
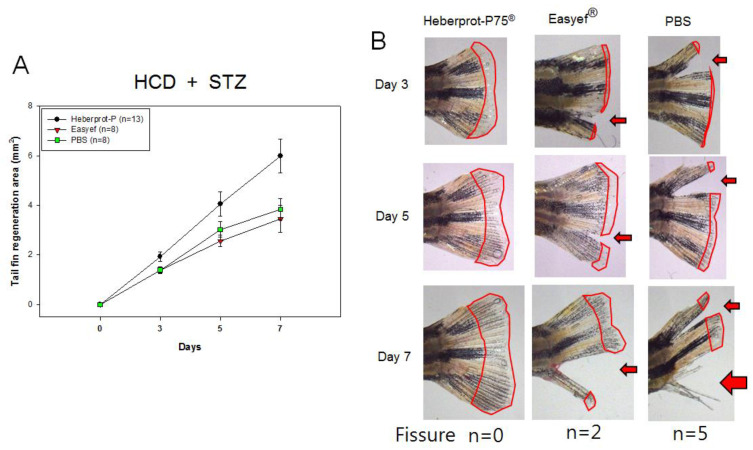
Comparison of the tissue regeneration speed (**A**) and morphology in tail fin area (**B**) under HCD consumption alone over seven days. The red arrow indicates the tail fissure. At bottom of photos, n indicates the number of zebrafish that showed fissure of tail fin on day 7. HCD, high cholesterol diet; STZ, streptozotocin.

**Figure 9 geriatrics-07-00045-f009:**
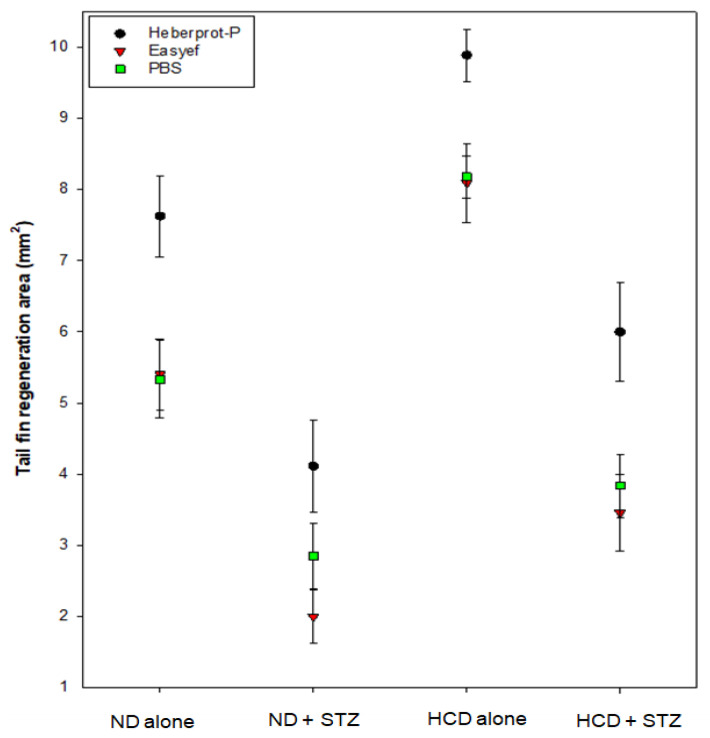
Tissue regeneration area at seven days post-injection depends on the diabetic condition induced by streptozotocin (STZ) injection and on hyperlipidemia by cholesterol feeding. ND, normal diet; HCD, high-cholesterol diet.

## Data Availability

The data used to support the findings of this study are available from the corresponding author upon reasonable request.

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
