# Peer review of "Efficacy Comparison Study of Human Epidermal Growth Factor (EGF) between Heberprot-P^®^ and Easyef^®^ in Adult Zebrafish and Embryo under Presence or Absence Combination of Diabetic Condition and Hyperlipidemia to Mimic Elderly Patients"

_geriatrics, 2022, doi:10.3390/geriatrics7020045_

Round 1
Reviewer 1 Report
The authors compare the in vivo efficacy of two EGF products in tissue regeneration and wound healing activity. Each product has different properties and vias of administration that was also tested in the zebrafish model. This model is interesting and very appropriate to evaluate the effect of pharmaceuticals such as in this work, moreover in diabetic and cholesterol supplementation conditions. It was clear that Heberprot-P75 had a better effect in wound healing and tissue regeneration that Easy even in diabetic or cholesterol supplementation. This was an important study particularly to evaluate a better choice of pharmaceutical in elderly patients.
Author Response
Dear reviewer,
Thank you for your helpful comments after careful reading of my manuscript.
I appreciate your valuable comments to improve the manuscript
We also absolutely agree with your opinion.
We added the aim of the research in Introduction as below
“The current study was designed to compare the in vivo efficacy of two EGF products in tissue regeneration and wound healing activity. Heberprot-P® and Easyef® were injected individually into the intraperitoneal space of adult zebrafish, in the absence or presence of both diabetic condition and cholesterol supplementation to mimic hyperlipidemia of elderly.”
“The in vitro and in vivo evaluations of the two EGF pharmaceuticals can provide useful informations about better choice for elderly patients.”
Reviewer 2 Report
The manuscript is interesting and well written, but contains some imperfections that need to be corrected.
In detail:
Introduction
The aim of the research must be presented in detail
Materials:
line 96
please be more specific - which samples?
lines 117-120
Please combine into one sentence to avoid repetition
line 328 - E. coli should be written in italics
Author Response
Dear reviewer,
Thank you for your helpful comments after careful reading of my manuscript.
I appreciate your valuable comments to improve the manuscript.
All suggestions from the reviewer are reflected in this revision.
The revised sentences are indicated in blue font.
The manuscript is interesting and well written, but contains some imperfections that need to be corrected.
In detail:
Introduction
The aim of the research must be presented in detail
Agreed. We also absolutely agree with your opinion.
We added the aim of the research in Introduction as below
“The current study was designed to compare the in vivo efficacy of two EGF products in tissue regeneration and wound healing activity. Heberprot-P® and Easyef® were injected individually into the intraperitoneal space of adult zebrafish, in the absence or presence of both diabetic condition and cholesterol supplementation to mimic hyperlipidemia of elderly. Zebrafish embryos were also treated with Heberprot-P® and Easyef® to compare the influence of water exposure and microinjection on developmental speed and morphology. Heberprot-P® and Easyef® were compared from the spectroscopic properties regarding intrinsic Trp fluorescence and collisional quenching as in vitro comparison. The in vitro and in vivo evaluations of the two EGF pharmaceuticals can provide useful informations about better choice for elderly patients.”
Materials:
line 96
please be more specific - which samples?
Agreed. We provided detail information as below.
“The equally diluted three samples, Heberprot-P75® , Easyef® , and methyl parabenzoic acid, were excited at 295 nm to avoid tyrosine fluorescence. The emission spectra were scanned from 190 to 900 nm at room temperature.”
lines 117-120
Please combine into one sentence to avoid repetition
Agreed. We combine the sentences to be concise.
“For the fin regeneration studies, zebrafish were anesthetized by submersion in 2-phenoxyethanol (Sigma P1126; St. Louis, MO) in system water (1:1000 dilution). Tail fins of the zebrafish, approximately 24 weeks old, were cut with a scalpel close to the proximal branch point of the dermal rays within the fin.”